# Mechanical forces control the valency of the malaria adhesin VAR2CSA by exposing cryptic glycan binding sites

Rita Roessner[1¤a], Nicholas Michelarakis[1¤b], Frauke Gräter[1,2], Camilo Aponte-Santamaría[1]*

**1** Molecular Biomechanics Group, Heidelberg Institute for Theoretical Studies, Heidelberg, Germany,
**2** Interdisciplinary Center for Scientific Computing, Heidelberg University, Heidelberg, Germany

¤a Current address: IBMM, University of Montpellier, CNRS, ENSCM, Montpellier, France
¤b Current address: Early Stage Pharmaceutical Development Biologicals, Boehringer Ingelheim Pharma GmbH & Co. KG, Biberach/Riss, Germany
* camilo.aponte@h-its.org

**Data Availability Statement:** Relevant data are within the manuscript and its Supporting information files. Scripts to setup the MD simulations and to analyze the output data, as well

## Abstract

Plasmodium falciparum (*Pf*) is responsible for the most lethal form of malaria. VAR2CSA is an adhesin protein expressed by this parasite at the membrane of infected erythrocytes for attachment to the placenta, leading to pregnancy-associated malaria. VAR2CSA is a large 355 kDa multidomain protein composed of nine extracellular domains, a transmembrane helix, and an intracellular domain. VAR2CSA binds to Chondroitin Sulphate A (CSA) of the proteoglycan matrix of the placenta. Shear flow, as the one occurring in blood, has been shown to enhance the (VAR2CSA-mediated) adhesion of *Pf*-infected erythrocytes on the CSA-matrix. However, the underlying molecular mechanism governing this enhancement has remained elusive. Here, we address this question by using equilibrium, force-probe, and docking-based molecular dynamics simulations. We subjected the VAR2CSA protein–CSA sugar complex to a force mimicking the tensile force exerted on this system due to the shear of the flowing blood. We show that upon this force exertion, VAR2CSA undergoes a large opening conformational transition before the CSA sugar chain dissociates from its main binding site. This preferential order of events is caused by the orientation of the molecule during elongation, as well as the strong electrostatic attraction of the sugar to the main protein binding site. Upon opening, two additional cryptic CSA binding sites get exposed and a functional dodecameric CSA molecule can be stably accommodated at these force-exposed positions. Thus, our results suggest that mechanical forces increase the avidity of VAR2CSA by turning it from a monovalent to a multivalent state. We propose this to be the molecular cause of the observed shear-enhanced adherence. Mechanical control of the valency of VAR2CSA is an intriguing hypothesis that can be tested experimentally and which is of relevance for the understanding of the malaria infection and for the development of anti placental-malaria vaccines targeting VAR2CSA.

as molecular dynamics simulation parameters and relevant input and output files, are available here: https://doi.org/10.11588/data/EZHMDU.

**Funding:** We are grateful for financial support by the Klaus Tschira Foundation (to all authors). The funders had no role in study design, data collection and analysis, decision to publish, or preparation of the manuscript.

**Competing interests:** The authors have declared that no competing interests exist.

## Author summary

Pregnancy associated malaria affects millions of children and women and causes over tens of thousands of deaths each year. Malaria is caused by infection with the Plasmodium parasite, *Plasmodium falciparum* being the deadliest one. Pregnancy-associated malaria is related to the adhesion of infected erythrocytes to the placenta. The adhesion is mediated by a parasite protein called VAR2CSA, which anchors infected erythrocytes to the chondroitin sulphate A (CSA) sugar matrix of the placenta. Accordingly, VAR2CSA is a promising anti-placental malaria vaccine target. Here, by using atomistic simulations, we investigated the molecular mechanism of adhesion mediated by this protein, so far unknown. We demonstrate that mechanical forces, like the ones experienced by VAR2CSA due to the shear of the flowing blood, open VAR2CSA into two structurally-functional domains, exposing two cryptic CSA binding sites apart from the already-exposed main site. Thus, we propose that mechanical forces control the number of CSA binding sites in VAR2CSA and that shear flows increase the adherence of infected erythrocytes to the placenta following this mechanism. This intriguing hypothesis is testable experimentally and is relevant for our understanding of the malaria infection and for the design of vaccines against pregnancy-associated malaria targeting VAR2CSA.

## Introduction

*Plasmodium Falciparum* (*Pf*) is responsible for the most virulent and deadliest form of malaria [1, 2]. During the blood stage of the infection, the *Pf* parasite infects, multiplies within, ruptures, and reinfects the erythrocytes of the host organism [3]. To achieve this, the parasite substantially modifies the morphology and content of the invaded erythrocytes [4]. In order to avoid the immune response of the host, i.e. the clearance of the infected red blood cells by the spleen, *Pf* sequesters itself in various organs by expressing a multitude of anchor proteins on their surface [5]. This family of proteins, known as Plasmodium falciparum Erythrocyte Membrane Protein 1 (PfEMP1), is encoded by approximately 60 var genes and used to mediate binding to the host's vascular system [6]. Amongst this adhesin family, VAR2CSA is the sole protein used by the infected erythrocytes for binding to the placental cells through an unusually low-sulphated form of chondroitin sulphate A (CSA) found in the placental intervillous space [7–11] (Fig 1, top panel).

Pregnancy-associated malaria (PAM) affected over eleven million pregnant women in 2020 [2]. Although the infection can be clinically silent, it often leads to maternal anaemia, severely impaired fetal growth, still birth, and spontaneous pregnancy loss. Over 200,000 infant deaths, 10,000 maternal deaths, and 900,000 low-weight births are attributed to PAM each year alone [7, 12, 13]. Women gain immunity to the disease through successive pregnancies, by acquiring specific antibodies against the VAR2CSA protein [14–16]. Nevertheless, pregnant women are shown to be more susceptible to malaria infection compared to non-pregnant women despite pre-existing immunity [7]. This is attributed to the formation of the placenta, which offers a new habitat for *Pf*-infected erythrocytes. Because it is precisely the protein that mediates the adhesion of *Pf*-infected erythrocytes and the CSA proteoglycan matrix of the placenta, VAR2CSA is a promising target for vaccine development against placental malaria [17]. In fact, there are two vaccine candidates in phase I/II clinical trials [18, 19].

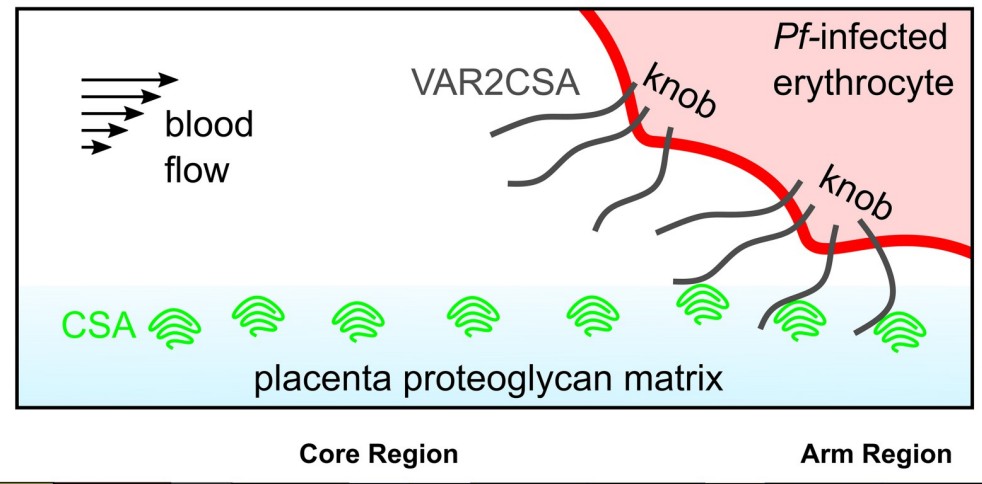

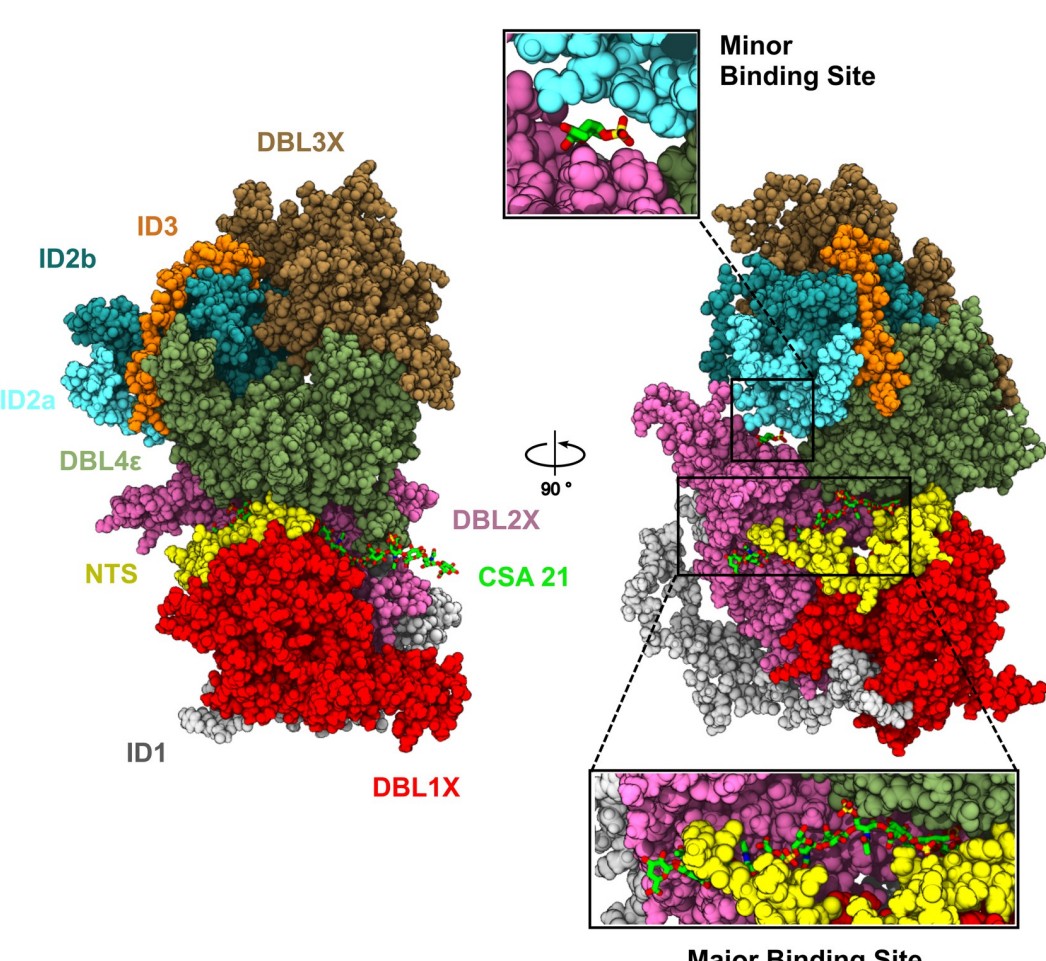

**Fig 1. VAR2CSA structure and function.** (Top) A schematic depicting the expression of VAR2CSA on a *P. Falciparum* (*Pf*) infected erythrocyte and its attachment to the intervillous CSA, in a blood vessel. (Middle) Sequence of protein domains (coloured squares) of the extracellular core and arm regions of VAR2CSA. (Bottom) Two views of the VAR2CSA core region (same colours as in the middle panel). The major and minor CSA binding sites are highlighted, with close-up views showing the bound CSA moieties (green sticks).

VAR2CSA is a multidomain protein composed of extracellular, transmembrane, and intra-cytoplasmic regions (Fig 1, top panel). VAR2CSA embeds itself in the membrane of the infected erythrocyte through a single membrane-spanning segment. The extracellular region comprises a short N-terminal segment (NTS) and six Duffy-like binding domains (DBL1X, DBL2X, DBL3X, DBL4ε, DBL5ε, and DBL6ε) which are relatively well conserved amongst parasites [20] (Fig 1, middle panel). They are connected by four complex, inter-domain regions (ID1, ID2a, ID2b, and ID3) which exhibit little homology amongst members of the PfEMP1 family [21, 22] (Fig 1, middle panel). Due to its large size of approximately 355 kDa, the structural information available until recently was very scarce, relying mostly on the structures of single domains [23–26].

CSA binds with high affinity and specificity to DBL2X and its flanking interdomain regions located at the N-terminal part of VAR2CSA [27, 28]. Structural information of VAR2CSA obtained by low-resolution small angle X-ray scattering (SAXS) revealed the overall shape of the full extracellular region as well as the minimal binding unit [28, 29]. SAXS data from reference [30] revealed that the extracellular region is divided into a compact core and a flexible and potentially extensible arm (Fig 1, middle panel). Furthermore, it was suggested that the core region contains two separate CSA-binding pores. However, only recently, the cryogenic electron microscopy (cryo-EM) structure of the NF54 VAR2CSA strain was elucidated [31]. The structure contained a CSA dodecamer sugar chain, i.e. the minimal functional binding unit [32, 33], bound to the previously-identified region between the DBL1X, NTS, and DBL2X domains, in the following referred to as the major binding site (Fig 1, bottom panel). Moreover, this structure displayed additional electron density that was attributed to a single CSA moiety, located in a groove between the DBL2X, ID2a, and DBL4ε domains, referred to here as the minor or cryptic binding site (Fig 1 bottom panel). Two apo structures of the VAR2CSA core region (FRC3 strain) [21] displayed high similarity with the holo CSA-bound structure [31]. Furthermore, molecular dynamics (MD) simulations confirmed a similar major CSA-sugar binding site as seen for the NF54 and the FCR3 strain [21]. Taken together, the cryo-EM and simulation data showed that the structural fold of the core region is highly conserved across different VAR2CSA strains and not significantly altered by CSA binding. However, a cryo-EM study of yet another VAR2CSA strain (i.e. 3D7) contradicts these results, suggesting large conformational changes upon sugar binding [34].

Cytoadhesion of *Pf*-infected erythrocytes occurs under shear stress caused by the blood flow. Parameters such as the distribution of PfEMP1 on the erythrocyte surface [35] as well as the receptor's density [36, 37] have been found to be critical for the kinetics of adhesion. For the particular case of VAR2CSA, Rieger and co-workers [36] demonstrated that shear stress enhanced the number of adherent *Pf*-infected erythrocytes expressing this adhesin. This result suggested the possibility of a catch-bond adhesion mechanism for VAR2CSA, in which the protein undergoes a conformational change upon the application of force and thereby increases its adherence. Catch bonds mediate the adhesion of *E. coli* bacteria to mannosylated surfaces, via the force-increased interaction of the fimbrial adhesin protein to its glycan partner [38]. Selectins are another example employing catch bonds during the rolling adhesion of leukocytes [39, 40]. Moreover, another PfEMP1 variant has also been suggested to exhibit a catch-bond mode of binding [41]. Catch bonds are contrary to slip bonds, in which the binding reduces when they are subjected to force [42].

In this work, we employ extensive equilibrium, force-probe, and docking-based MD simulations to elucidate the, still unknown, molecular mechanism that leads to increased VAR2CSA–CSA adhesion under shear stress.

## Methods

### VAR2CSA model construction and equilibrium MD simulations

The VAR2CSA NF54 strain was considered in this work. A refined and complete model of the core domain (residue IDs 1–1955) was obtained using existing cryo-EM structures of the full VAR2CSA ectodomain (PDB ID. 7JGH, 7JGD, and 7JGE [31], and 7B52 [21]), as well as X-ray structures of individual domains (DBL3X: PDB ID. 3CML and 3CPZ [24], and 3BQI, 3BQK, and 3BQL [23]) and of the DBL3X-DBL4$\epsilon$ complex (PDB ID. 4P1T [26]). All these structures were combined into a modelling protocol with Modeller 9.18 [43] to complete missing residues and loops that were absent in the full ectodomain structures. The modelled segments comprising residues 407–485 and 529–556 were rotated manually after the modelling procedure to prevent their wrapping around the protein. The CSA dodecamer found in the major CSA binding site of the NF54 structure was extended to 21 subunits using PyMOL [44]. Subsequently, the system was energy minimized with the steepest descent algorithm applying harmonic restraints on the atoms which were observed in the NF54 structure (harmonic elastic constant was 5000 kJ/mol/nm$^2$ and reference positions for the harmonic potentials were taken from the NF54 [31] structure). The resulting model displayed a root mean square deviation (RMSD) of 0.23 Å from the NF54 structure (PDB ID. 7JGH [31]) and of 1.43 Å from the FCR3 strain structure (PDB ID. 7B52 [21]). The final model was then equilibrated through the use of equilibrium MD simulations, following a protocol of progressive restraint relaxation as explained as follows.

After energy minimization, the conformation of the loops (residues 407–485 and 529–556) was refined with Langevin stochastic dynamics in vacuum for 100 ns. The rest of the system was kept frozen at a temperature of 500 K with a friction time constant of 0.002 ps. The protein was subsequently solvated with water and 150 mM NaCl in a dodecahedron box with an excess of chloride ions to neutralize the net charge of the protein. The resulting system consisted of approximately 0.7 million atoms. The solvated system was energy minimized and simulated under NVT-ensemble conditions for 500 ps at a temperature of 310 K. All heavy atoms were position-restrained with a force constant of 1000 kJ/mol/nm$^2$. In a second stage, the system was equilibrated in an NPT ensemble, under the same conditions, for 1 ns. This step was repeated with all non-modelled atoms being restrained with a force constant of 1000 kJ/mol/nm$^2$. The system was further equilibrated in an NPT ensemble for 1 ns with position restraints on the backbone atoms. The force constant in this instant was 100 kJ/mol/nm$^2$. Finally, the system was simulated under the same conditions for 200 ns, without any backbone restraints. Ten replicas were simulated for a total of 2 $\mu$s of cumulative simulation time.

### Force probe MD simulations

Two frames of each replica from the equilibrium simulations described above were extracted and used as starting points for 20 pulling simulations (one frame was taken at the end of the simulation and a second frame 30 ns before). Each conformation was subsequently re-solvated in a cubic box with water and ions (as in the simulations above) resulting in a system of approximately 1 million atoms. The solvated systems were independently thermalized via a 500 ps NVT simulation, maintaining protein backbone atoms position-restrained with a force constant of 100 kJ/mol/nm$^2$. The force application points were selected to mimic the tensile force experienced by the core domain of VAR2CSA under blood flow, namely, the C-terminal anchoring point of VAR2CSA in the erythrocyte membrane and the 1'-O-glycosidic linkage of glucuronic acid to the placental proteoglycans [45]. Accordingly, the first residue of the bound CSA molecule (numbering according to Fig 2A) and the C-terminal

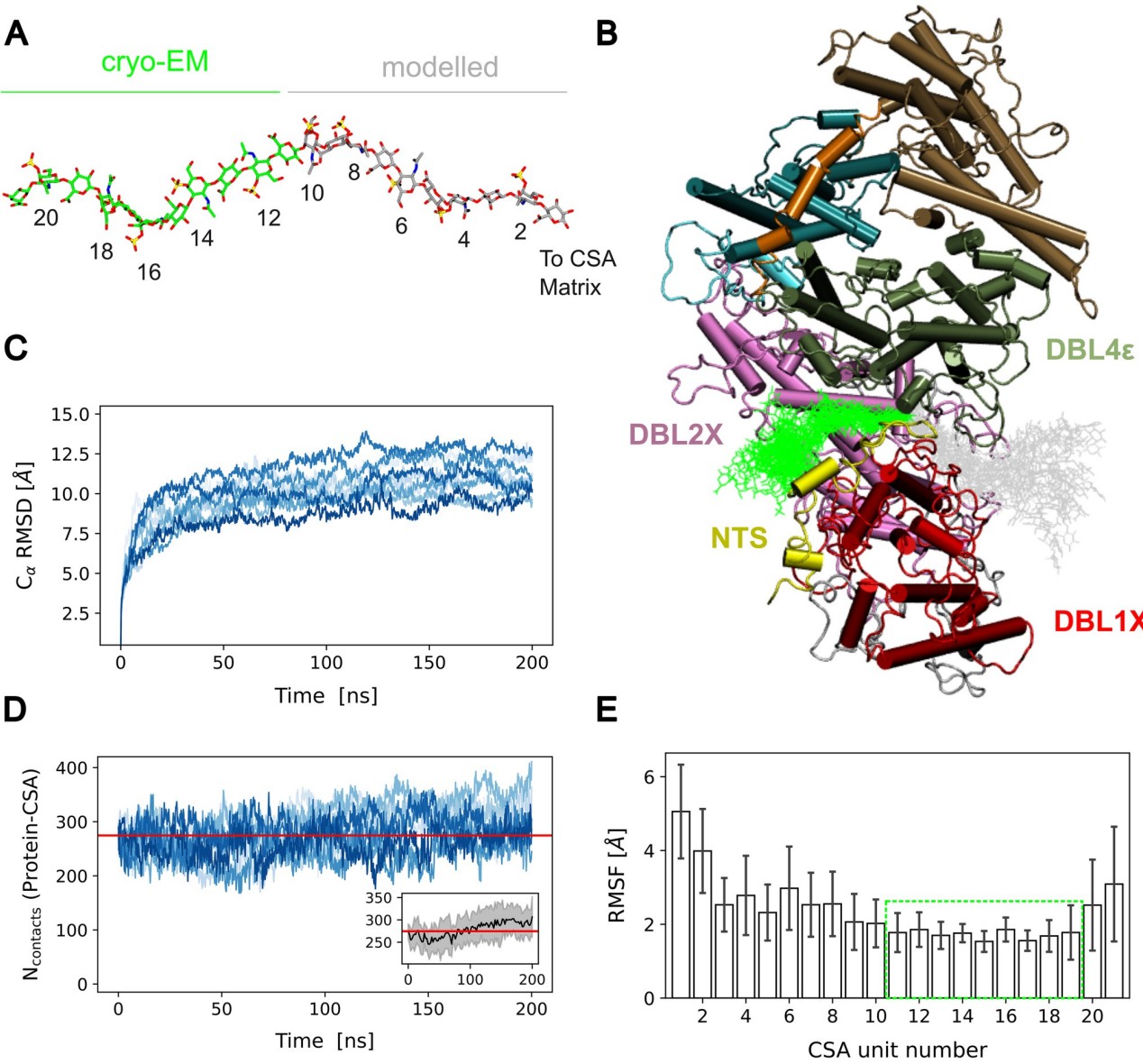

**Fig 2. MD Equilibration of VAR2CSA with a CSA 21mer sugar chain bound to it.** (A) The CSA molecule is presented in stick representation (green: cryo-EM resolved part, PDB ID. 7JGH [31]; grey: modelled part). The numbering of the monosaccharides increases with respect to the terminus that points to the CSA matrix. (B) Example of the conformations sampled by the CSA 21-mer in one of the 200 ns MD simulations when the molecule was bound to the major binding site of the core region of VAR2CSA. The sugar is shown as sticks (coloured as in A) and the protein is displayed in cartoon representation (coloured according to Fig 1). (C) Cα root mean square deviation (RMSD) from the initial conformation of the VAR2CSA core region, for n = 10 independent replicas (differently coloured lines). (D) Time traces of the number of contacts between VAR2CSA and the CSA chain which was bound to the major binding site (n = 10 independent replicas). Inset: Average ± standard deviation of the number of contacts as a function of time. To guide the eye, the red lines depict the initial number of contacts (in average). (E) Root mean square fluctuation (RMSF) of each CSA monosaccharide (average ± standard deviation, n = 10). The Green Box highlights the part of the sugar resolved by cryo-EM, which exhibited a relatively low conformational flexibility.

methionine residue of the core protein were subjected to harmonic spring potentials, with a force constant of 1000 kJ/mol/nm$^2$ and moving with a constant velocity of 0.1 m/s in opposite directions. Each replica was simulated for 100 ns for a cumulative force-probe simulation time of 2 µs.

## CSA dodecamer docking at cryptic binding sites and MD relaxation

To understand whether force-exposed cryptic binding sites have the ability to accommodate a functional CSA molecule, we docked a CSA dodecamer to the open VAR2CSA molecule using the HADDOCK web server [46]. We selected this size of saccharide since a dodecamer is the minimum length needed for CSA binding in the major CSA binding site [32, 33]. The dodecamer was treated as "ligand" while the VAR2CSA together with the 21mer sugar chain bound to the major binding site were considered as the "receptor" during the docking calculation. The structure of the dodecamer (ligand) was taken from the cryo-EM structure (PDB ID. 7JGH [31]). For the protein with the bound sugar 21mer (receptor), we selected an open conformation with the N-terminal and C-terminal subregions separated by a distance of 9 nm (see Fig 3C) from each replica of the pulling simulations, resulting in 20 initial input receptor configurations. The dodecamer was docked at two different locations of the protein: one corresponding to the minor binding site, which locates at the interface between the DBL2X and the ID2a domain, and the other corresponding to the DBL4$\epsilon$ domain portion of the major binding site that has been observed to become exposed in the force probe simulations. Accordingly, based on the cryo-EM holo structure of VAR2CSA (PDB ID. 7JGH [31]), the active residues directly involved in the interaction were assumed to be T910, Y911, T912, T913, H949, K952, D968, and K970 for the minor binding site and H1782, I1783, G1784, I1785, I1878, M1879, E1880, K1889, R1890, N1896, N1898, and Y1899 for the DBL4$\epsilon$ site. In addition, the first five sugars of the dodecamer were treated as the active region involved in the interaction. No passive residues were considered in the docking calculation. The number of structures for rigid body docking was set to 10,000 while the number of structures for both semi-flexible refinement and explicit solvent refinement was 400. Default values were considered for all the other docking parameters. HADDOCK clustered the docking conformations into eight different groups. A representative pose of each cluster was selected. Each docked conformation was subsequently solvated and equilibrated in the same manner as for the force-probe simulations. The stability of the docked dodecamer was monitored in simulations from 55 to 100 ns. To avoid re-closing of the VAR2CSA subregions, we applied a weak flat-bottom potential to the first residue of the CSA molecule and the C-terminal residue of the core protein. This potential was larger than zero when the distance between the pulled groups was below 13 nm and zero otherwise. The "flat-bottom-high" pulling coordinate option in GROMACS was used for this purpose. The elastic constant of this potential was 1000 kJ/mol/nm$^2$.

## Simulation algorithms and parameters

Through all the simulations described in this work the temperature was maintained at 310K, through the use of the Nose-Hoover thermostat [47, 48], coupling the protein and the rest of the system separately to the thermostat, and using a coupling time constant of 1 ps. The pressure was kept constant at 1 atm by means of the Parrinello-Rahman barostat [49, 50] (coupling time constant of 5 ps). The GROMACS 2019.4 [51] molecular dynamics package was used for the simulations, with an integration time step of 2 fs. The CHARMM36m forcefield [52] was used for the protein, the CHARMM-modified TIP3 model [53] for the water molecules, and CHARMM parameters for the CSA sugar and the NaCl ions. The CHARMM-GUI [54–56] web server was used for the generation of the sugar forcefield parameters and the GROMACS gmx tools for the remaining force-field terms. Electrostatic interactions were calculated by using the Particle Mesh Ewald method [57, 58] in the direct space within a cut-off distance of 1.2 nm and in the reciprocal space beyond that cut-off. Electrostatics were treated with reaction field in the in-vacuum loop relaxation step with a dielectric constant of 1. Short-range interactions were modelled through a Lennard Jones potential within a cut-off distance of 1.2

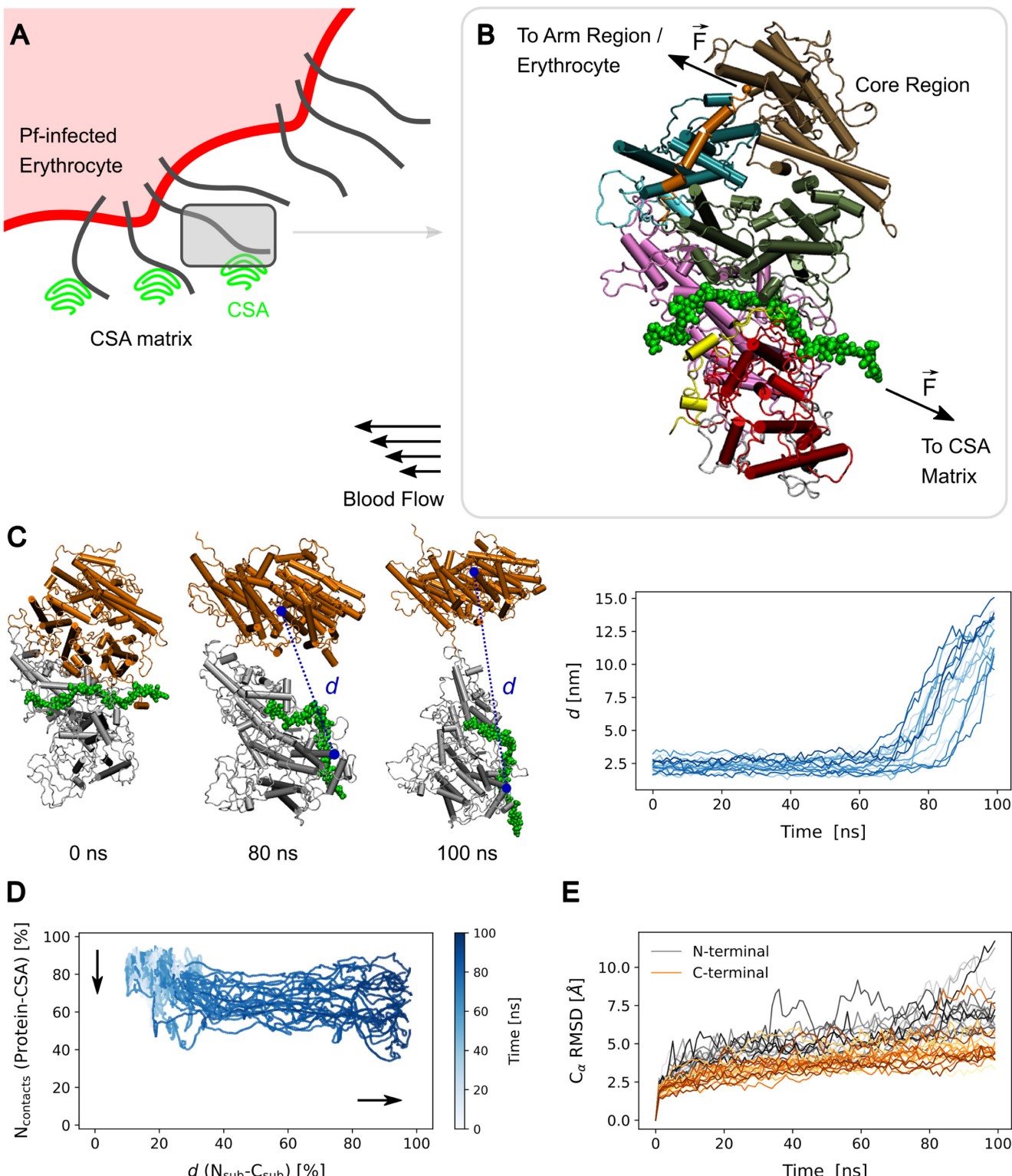

**Fig 3. Force-induced opening of the VAR2CSA core region.** (A) The adhesion of a *Pf*-infected erythrocyte (red) to CSA molecules of the intervillous matrix (green) is mediated by VAR2CSA (black lines). The shear of flowing blood drags the erythrocyte, thus excerting tensile force on VAR2CSA. (B) To mimic this tension, harmonic forces $\vec{F}$ were exerted on the VAR2CSA core–CSA complex in force-probe MD simulations (VAR2CSA core: cartoon, coloured according to Fig 1, CSA: green spheres). Force was applied to the first monomer of the CSA chain (anchor point to the CSA matrix) and at the C-terminal methionine residue of the core region (linked to the infected erythrocyte by the arm region). (C) Left: example of opening of the core region. The

N-terminal subregion (domains NTS, DBL1X, ID1 and DBL2X, i.e. residues 1–964) is coloured in gray while the C-terminal one (ID2a, ID2b, DBL3X, DBL4$\epsilon$, and ID3, i.e. 965–1954) is shown in orange. The CSA chain is depicted in green. Right: distance $d$ between the residues 120 and 1663 (dashed blue line on the left snapshots) quantified the degree of opening. $d$ is shown for all 20 independent simulation replicas. See also S1 Movie. (D) The number of VAR2CSA–CSA contacts ($N_{contacts}$) is presented as a function of the distance ($d$) and time (colour) for the 20 simulation replicas. $N_{contacts}$ inversely relates to the dissociation of the CSA 21mer from the major binding site (vertical arrow) whereas the $d$ corresponds to the opening of the interface between subregions (horizontal arrow). $N_{contacts}$ is proportional to the initial number of contacts and $d$ relative to the maximum opening distance sampled. Accordingly, $N_{contacts}$ = 0% would correspond to a fully dissociated state (never observed) and $d$ = 100% to the conformation with the maximum observed amount of opening. Linear regression of these curves yielded a slope of -0.28±0.17 (average ± standard deviation, n = 20). A slope >-1 implies that opening occurs before than dissociation and <-1 the opposite. The obtained slopes illustrate the trend of the low amount of sugar contacts lost during opening. (E) C$\alpha$ root mean square deviation (RMSD) of the N-terminal- (grey) and C-terminal subregion (orange) as a function of time (n = 20).

nm. Neighbour atoms were considered through the Verlet buffer scheme [59]. The Lennard Jones force was shifted between 1 nm and 1.2 nm. Bonds involving hydrogen atoms were constrained by using the LINCS algorithm [60] and both bonds and angles of water molecules were constrained via SETTLE [61].

## Simulation analysis

The simulation analysis was performed with GROMACS tools [51] and MDAnalysis 2.3 [62, 63].

**Root mean square deviation and fluctuations.**   Root-mean-square deviations (RMSD) of atomic distances and root-mean-square fluctuations (RMSF) of atomic positions were determined from the equilbrium MD simulations with the MDAnalysis.analysis.rms module [64, 65]. For the RMSF calculation, the last 100 ns of each simulation replica were considered.

**Contact analysis.**   Contact analysis was performed using the GROMACS tool *gmx mindist* whereby two atoms were in contact when their minimum distance was smaller than 4.5 Å. To evaluate the stability of the VAR2CSA/CSA complex in the equilibrium MD simulations, we computed the number of contacts between the protein and CSA over the course of the trajectories. We monitored the opening of the VAR2CSA core in the force probe simulations by calculating the number of VAR2CSA–CSA contacts as a function of the distance between the N-terminal- and C-terminal subregion and as a function of time. We also computed the number of contacts between the CSA dodecamer docked to the DBL4$\epsilon$- and DBL2X cryptic binding site to evaluate its interaction with the VAR2CSA/CSA complex.

**Opening of VAR2CSA core.**   The opening of the VAR2CSA core was additionally monitored over the course of the force probe simulations by measuring the distance between the residues 120 and 1663 of the N-terminal- and C-terminal subregion, respectively.

**Force distribution analysis.**   The pair-wise force $F_{ij}$ between the pair of residues $(i, j)$ was computed by using the force distribution analysis (FDA) tool implemented by Costescu and Gräter [66] (version 2.10.2). The pair-wise force difference $\Delta F_{ij} = \langle F_{ij}(\text{open}) \rangle - \langle F_{ij}(\text{closed}) \rangle$ was computed, with "open" ("closed") corresponding to the state after (before) occurrence of the opening of the VAR2CSA core region. $\langle \rangle$ denotes an ensemble average. In practice, time averages in a 2 ns time window at 20 ns ("closed") and at 90 ns ("open") were computed, separately for each pulling simulation replica. The ensemble average was subsequently obtained by averaging these time averages over at least $r$ = 15 replicas. To give an idea of the statistical significance of the change, the following $z$ score function was evaluated $z =: \Delta F_{ij} / [\sigma_r^2(F_{ij}(\text{open})) + \sigma_r^2(F_{ij}(\text{closed}))]$, with $\sigma_r^2$ denoting standard deviation squared over the $r$ replicas. Pairs at different $z$ threshold values were shown. In addition, pairs showing a very small change, i.e. smaller than 1 pN, were excluded. FDA was carried out in two different situations: one, when the pairs were significantly separated during the opening process, and, two, when they were not. We called the pairs "disrupted pairs" in the first case, because the FDA

tool excluded their interaction. Accordingly, we assumed that $\langle F_{ij}(\text{open})\rangle$ and $\sigma_r^2(F_{ij}(\text{open}))$ were equal to zero in this case. In the second case, we refer to "No pair disruption" because pairs were short ranged and thus FDA computed their interaction. In such a case we took the values of $F_{ij}(open)$ retrieved by the FDA tool.

**Orientation analysis.**   To shed light on the mechanism of CSA detachment and VAR2CSA opening, we evaluated the orientation of CSA bound to the major binding site as well as the orientation of the subregion interface with respect to the pulling axis over the course of the force-probe MD simulations. The vector between the points of force application on the VAR2CSA and the pulled CSA molecule defined the pulling direction. A suitable third point, consisting of CSA-residue 11 and of protein residue F1744, defined the orientation angles $\theta$ (CSA detachment) and $\varphi$ (VAR2CSA opening), respectively.

**Potential energy of VAR2CSA–CSA and subregion interaction.**   The short-range Lennard-Jones- and Coloumb contributions of the interaction potential between VAR2CSA and CSA as well as between the N-terminal- and C-terminal subregion were computed using the GROMACS *gmx mdrun* utility.

**Hydrogen bonds.**   The number of hydrogen bonds between each amino acid of the major binding site and the CSA sugar chain was extracted from the equilibrium MD simulations. The probability of formation of a hydrogen bond was estimated as the number of frames the amino acid formed at least one bond, divided by the total number of frames. For the calculations, all equilibrium MD replicas (n = 10) were considered, discarding the first 100 ns of each of them, a time that was accounted as equilibration time. The hydrogen bonds were extracted using the *gmx hbond* tool. Hydrogen bonds involving the side chains of charged amino acids were labelled as ionic pairs.

**Sugar solvent accessible surface area (SASA) at cryptic binding sites.**   We computed the SASA of the DBL4ε binding site (residues N1782, K1787, K1889, R1890, N1896, N1898) and the minor binding site (residues K797, K905, H949, K952, K958, K1052) using the GROMACS *gmx sasa* over the course of the force probe simulation to monitor their force-induced exposure. To take into account the size of the sugar chain, a solvent probe radius of 0.5 nm was used for this calculation.

The molecular visualization packages VMD [67] and PyMOL [44] were used to render the snapshots of the system as described above.

## Results

### A refined model of the VAR2CSA core region in complex with a CSA dodecamer

We first generated a complete model of the VAR2CSA NF54 strain core region (ranging from the NTS to the ID3 domain) in complex with a CSA 21mer bound to the major binding site (Fig 1). The model is based (mainly) on the recently determined cryo-EM structures of this strain [31] and the VAR2CSA FRC3 strain [21]. We considered the CSA dodecamer that was observed in the cryo-EM structure and prolonged it by 10 monomers, to be able later to properly examine the force-response of the protein–sugar complex (Fig 2A). We then relaxed the model by conducting n = 10 independent equilibrium MD simulations of 200 ns each, for a cumulative simulation time of 2 $\mu$s (Fig 2B). Over the course of the simulations, the system stabilized within 150 ns at conformations that deviated by no more than 15 Å (carbon-alpha root mean square deviation, RMSD) from the initial conformation (Fig 2B and 2C). We find these are reasonable RMSDs given the large size of the protein and the abundance of flexible loops. In addition, we computed the RMSD for each domain separately (S1 Fig). As expected, the DBL domains were the most stable ones, deviating less from the initial conformation (with

RMSDs below 10 Å). The ID domains and the NTS domain displayed larger deviations, although ID2a showed RMSDs comparable to those of the DBL domains. In any case, the RMSD converged and fluctuated around constant values in almost all cases, particularly, for the domains engaging in interactions with the CSA sugar chains, i.e. NTS, DBL2X, DBL4e, and ID2a. Moreover, to check the motions between pairs of domains, we computed the inter-domain number of contacts (S2 Fig). The figure highlights the overall preservation of the initial number of contacts while giving clues on specific pairs which displayed a low variability at the interface (such as the DBL1X–DBL2X or the DBL3e–DBL4e pairs) or cases in which the variability was higher (e.g. NTS–DBL1X). We also inspected, the number of contacts between the CSA sugar and the VAR2CSA protein which fluctuated around the initial value (Fig 2D). This quantity displayed a slight increase which we attribute to the larger flexibility of the modelled part of the chain (see next). Nevertheless, the drift was rather small compared to the variation in the number of contacts across different simulation replicas and with the initial value (Fig 2D, inset). This result indicates that the sugar stayed stably-bound to the protein over the course of the simulations. Each monosaccharide displayed structural fluctuations of less than 5 Å (Fig 2E), which are similar to the fluctuations observed in a previous study of the FCR3 strain [21]. As expected, the part of the sugar that was observed in the cryo-EM structure displayed slightly lower structural fluctuations than the modelled part. Consequently, the MD structural relaxation generated a refined ensemble of conformations suitable to investigate the mechanical response of the VAR2CSA-CSA complex.

## Force opens the core region of tethered VAR2CSA

In order to understand the force-response of the VAR2CSA-CSA complex, we performed force-probe MD simulations. We applied force in a manner that mimics the tension the protein is subjected to, when it adheres to a *Pf*-infected erythrocyte to the CSA proteoglycan matrix under the action of a shear flow (Fig 3A). Under these conditions, VAR2CSA is expected to be pulled away from the tethering CSA sugar chain by the sheared *Pf*-infected erythrocyte. We considered the region of the protein which contains the main CSA binding sites, namely, the core region (Figs 1 and 3B). We did not consider the "arm" region, consisting of domains DBL5ε and DBL6ε, because it is not in direct contact with the CSA sugars [21, 30, 31] (Fig 3B). Accordingly, we applied a harmonic force on the C-terminal methionine residue of the core region (residue ID. 1953), assuming that force transmitted there by the arm region. An opposing harmonic force was applied to the first monosaccharide of the bound CSA 21mer, i.e. the point that tethers VAR2CSA to CSA (Fig 3B). The two pulled points were separated in multiple force-probe MD simulations (n = 20, see details in the methods). Upon application of force, we observed a substantial opening of the core region: the N-terminal subregion composed of the domains NTS, DBL1X, ID1 and DBL2X (shown in gray) shifted away from the C-terminal subregion constituted by ID2a, ID2b, DBL3X, DBL4ε, and ID3 (depicted in orange) (Fig 3C and S1 Movie). Another possibility could have been that force induced the detachment of the sugar from the major binding site before the core region opened. This possibility was ruled out after plotting the number of contacts between the protein and the sugar versus the distance between regions (Fig 3D). As the simulation time progressed, we observed that the system followed preferential paths in which opening always occurred first. The distance between subregions increased to a larger extent than the protein-sugar contacts decreased. In fact, upon opening, the number of contacts never reached zero (indicative of dissociation) but instead remained at least at 40% of the initial value (Fig 3D). Linear regression of this data further highlights the small reduction in the number of contacts upon opening (Fig 3D, caption). Note that the fold of these two subregions was largely preserved under force

application, apart from a residual unfolding of the loose C-terminus (Fig 3E). This is mainly attributed to the large number of disulfide bonds that helped VAR2CSA maintain its structural integrity. Thus, tensile force opens the core region of tethered VAR2CSA and separates it into two almost structurally-intact subregions.

To pinpoint force-bearing residues pairs resisting the protein opening, we inspected the redistribution of mechanical stress in the VAR2CSA protein upon opening. As a way to quantify this, we used force distribution analysis. We retrieved the pair-wise force inside the protein when it was in the open state, relative to the force in the closed state. The pair-wise force drastically changed for several residue pairs, indicating a large redistribution of mechanical stress among them (Fig 4). We distinguished between pairs whose interaction was largely disrupted (Fig 4A) from pairs whose interaction was not diminished (Fig 4B). In the former case, as somewhat expected, residues located at the interface of the two subregions were most notoriously affected. In particular, a large set of ionic or hydrogen-bond interactions between the DBL2X and DBL4X domains were compromised (Fig 4A). In the latter case, changes occurred more locally, i.e. separately in each subdomain (Fig 4B), and they were overall statistically less pronounced (see pairs only distinguishable for small $z$ scores in S3 Fig). This corroborates the mechanical stability of the individual domains, which undergo very little deformation during rupture, as also observed structurally with the per-domain and per-subregion RMSDs (see Figs 3E and S1). Not surprisingly, in both cases, the unfolded C-terminal segment displayed significant changes in pair-wise force. Consequently, this analysis reveals a specific cluster of

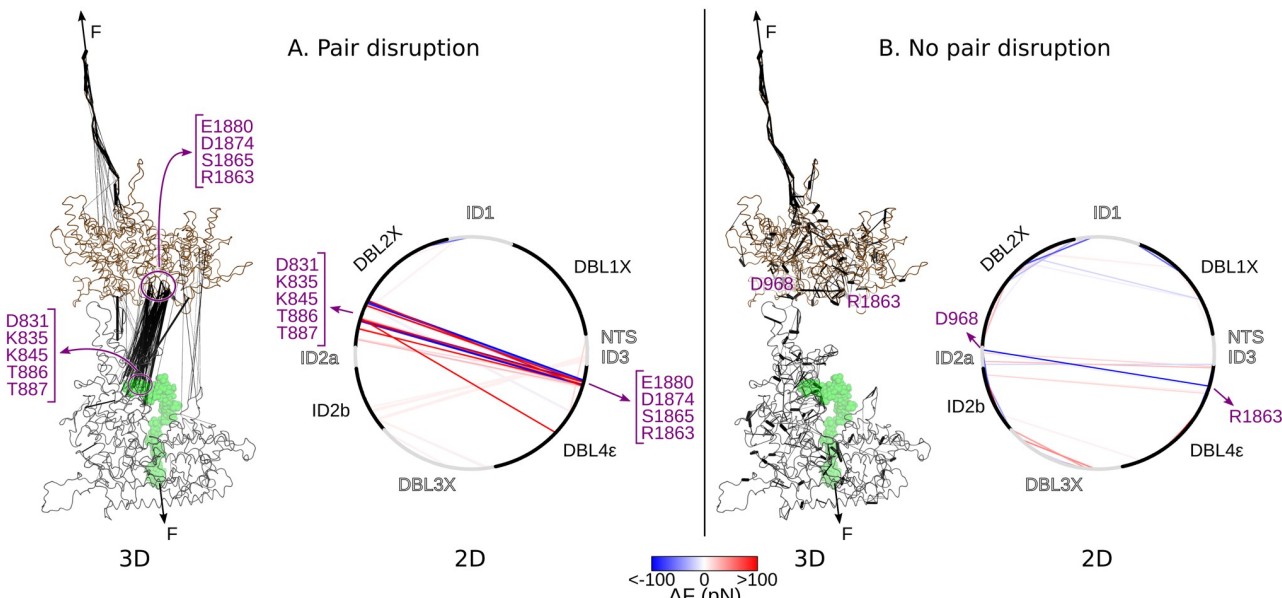

**Fig 4. Force distribution analysis identifies key residue pairs disrupted during opening.** Shown is the change in pairwise force $\Delta F_{ij} = \langle F_{ij}(\text{open}) \rangle - \langle F_{ij}(\text{closed}) \rangle$ for the pair of residues $(i, j)$. "open" ("closed") corresponds to the state after (before) opening of the VAR2CSA core region. $\Delta F$ is depicted as lines connecting the residues $(i, j)$ in either a 3D or 2D representation. Two situations were considered: when the interaction between the pair residues was completely disrupted during opening (A) and when it was not (B). In the 3D representations, VAR2CSA is shown as cartoon in one of its typical open conformations (N-terminal subregion: grey and C-terminal subregion: brown). The CSA chain is depicted with spheres (green). The thickness of the black lines is proportional to $|\Delta F|$. The points of application of force are indicated with the black arrows. In the 2D representations, the VAR2CSA amino acid sequence spans a circumference, starting with the NTS N-terminal domain and ending with the C-terminal ID3 domain. The intercalated black and gray arc segments correspond to the portion of each domain in the sequence. Here, $\Delta F$ is colored according to the color bar shown at the bottom. Accordingly, pairs that showed a pair-wise force higher in the open state are depicted in red, whereas pairs for which the force was higher in the closed state are indicated in blue. Only pairs displaying a statistically significant change are shown: score $z > 0.75$ in A and $z > 0.5$ in B (see pairs for other $z$ values in S3 Fig) and $|\Delta F| > 1$ pN. Residues displaying significant changes are highlighted in purple.

interdomain interactions, which are mechanically affected by the opening of VAR2CSA, and otherwise rather minor stress redistribution through the protein upon opening.

## Orientation of VAR2CSA and sugar–protein electrostatic interactions explain VAR2CSA opening preference over sugar dissociation

We next shed light on the mechanism behind the preferential opening of the core domain prior to detachment of the sugar from the major binding site. Because the pulling direction influences the force response of biomolecules [68–74], we first checked the effect of the orientation of VAR2CSA with respect to the pulling axis. We calculated the angles formed by the tethered sugar and the opened interface with respect to the pulling axis, denoted as $\theta$ and $\varphi$, respectively (Fig 5A). We observe that the CSA sugar inclined with respect to the pulling axes forming a $\theta$ angle of 20±12˚ (average ± standard deviation) (Fig 5B). The interface between the N- and the C-terminal subregions, in contrast, adopted a more perpendicular orientation with respect to the pulling axis ($\varphi$ = 51±11˚, average± standard deviation, Fig 5B). These numbers suggest that the orientation of the molecule along the line of tension causes the interface of the cryptic binding site to open in "zipping" mode, whereas the detachment of the CSA molecule from the major binding site follows a mechanically more resistant "shearing" motion.

In addition to the orientation, we checked whether the interaction strength between the interfaces subjected to force had an influence on the opening behaviour. To achieve this, we extracted the potential interaction energy between the CSA chain and the major binding site from the equilibrium MD simulations and between the two subregions (Fig 5C). Note that the potential energies only allow an approximate relative analysis of the energetics. Free energies would be required for a full quantitative picture. The difference in contact area resulted in a broader and overall stronger short-range Lennard-Jones contribution to the interaction between the subregions compared to the CSA-VAR2CSA interaction (Fig 5C, left panel). In contrast, the strong electrostatic interaction of the (negatively-charged) CSA sugar with the (positively-charged) major binding site was notorious in the Coulomb contribution compared to the subregion interface (Fig 5C, middle panel). This contribution is mainly attributed to the multiple and prevalent hydrogen bonds and ionic pairs that were formed between various amino acids in the major binding site and the CSA chain (Fig 5D). In fact, these interactions led to a stronger total potential interaction energy between the CSA sugar and the major binding site than that between subregions (Fig 5C, right panel). Thus, the sugar–protein electrostatic interactions also contribute to the opening of the interface while the CSA chain remains attached to the major binding site.

## Force causes the exposure of two cryptic CSA binding sites

We finally investigated the functional consequences of the opening of the VAR2CSA core due to the application of force. Remarkably, during the force-induced opening of the core region, the exposure of the DBL4ε residues that were engaged in direct interactions with the sugar chain at the major binding site increased. (Fig 6A, left). Note that despite the loss of such interactions, the CSA 21mer remained bound to the major binding site, more specifically to the DBL2X part of it (Fig 3D). In addition, the minor binding site at the groove of the DBL2X, ID2a, and DBL4ε domains increased its accessibility upon opening (Fig 6A, right). The exposure in this case was less notorious than that for the DBL4ε site, because this site was already partly exposed before the application of force.

A CSA chain could potentially be accommodated at these two newly-exposed sites. We tested this hypothesis by docking a CSA dodecamer, the minimal functional binding unit [32, 33], separately in either of such positions. We allowed the docked structures to relax in

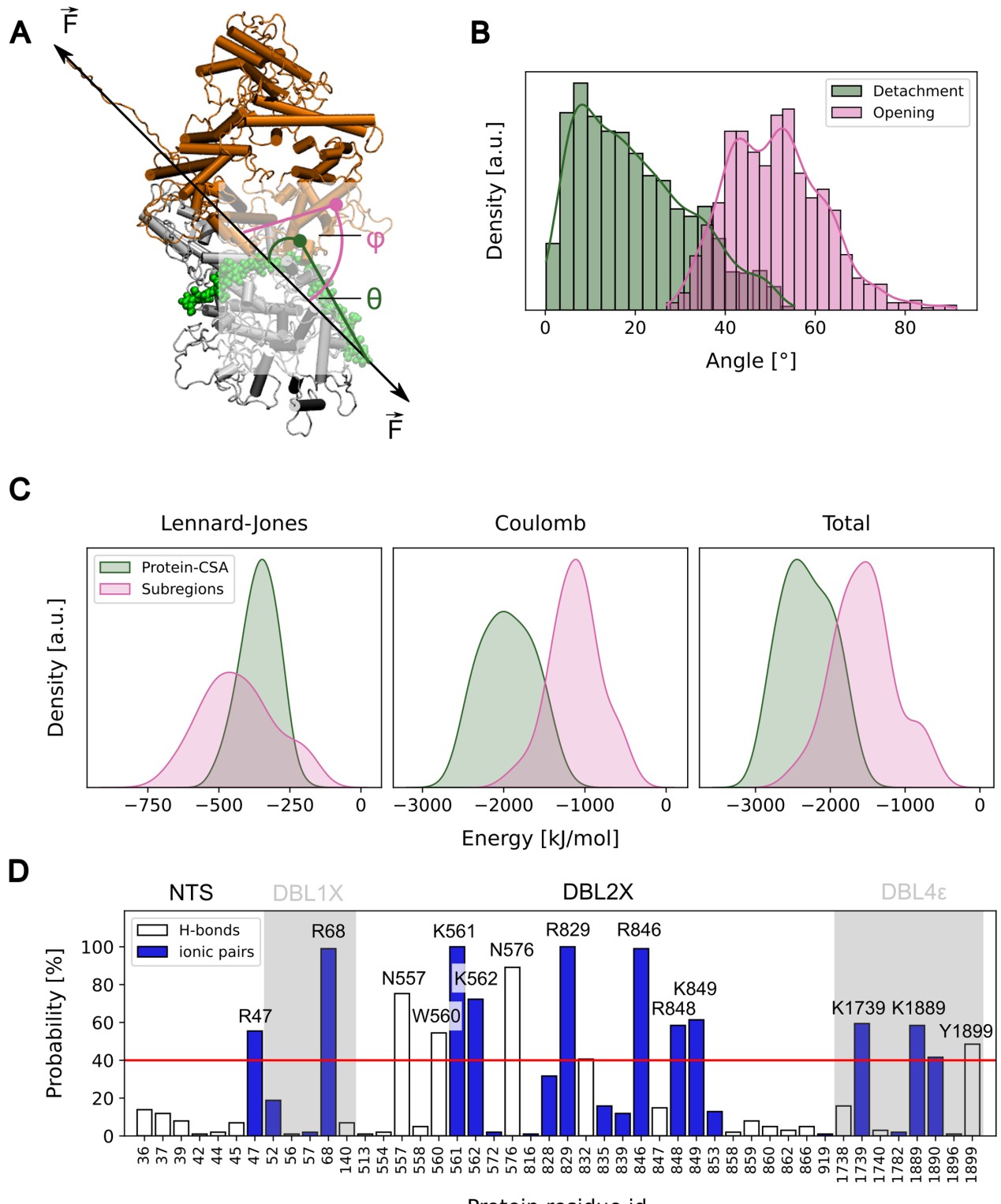

**Fig 5. Pulling orientation of VAR2CSA and protein–sugar electrostatic interactions can explain the opening of the tethered VAR2CSA core region.** (A) Cartoon showing the two angles, $\theta$ and $\varphi$, used for the exploration of the CSA detachment and opening mechanisms, respectively. $\theta$ is the angle formed by the CSA chain bound at the major binding site with respect to the pulling axis (green). $\varphi$ is the angle along the subregion interface with respect to the pulling axis (pink) (B) Histogram showing the distribution of $\theta$ (green) and $\varphi$ (pink) angles calculated over the 20 force-probe simulation repeats until the subregions were entirely separated. (C) The potential energy between the subregions that opened under force

(Subregions: pink) and between the CSA chain and the major binding site (protein–CSA: green) were computed from the equilibrium MD simulations. Normalized distributions of the short-range Lennard-Jones (left) and Coulomb (middle) contributions, as well as the distribution of the sum of them (right), are presented. (D) Probability of formation of hydrogen bonds between each amino acid of the major binding site and the CSA sugar (white bars). Probability of formation of ionic pairs, i.e. hydrogen bonds between the positively-charged side chains of Arginine, Lysine and Histidine, and the negatively-charged carboxylate and sulfate groups of the CSA chain is also shown (blue bars). Residues that displayed a probability higher than 40% are labeled. The amino acids are separated according to the domain they belong to.

simulations of several tens of ns (from 55 to 100 ns) while keeping the protein in the open state (see Methods). During this process, the CSA molecule remained stably bound at either of these two sites (Fig 6B). A quantitative indication of this is the stable fluctuations around a constant value in the number of CSA sugar–protein contacts, i.e. no loss of such contacts over time (Fig 6C). Due to its electrostatic nature, these contacts often occurred between the negatively-charged moieties of the CSA chain and positively-charged residues of the protein (Fig 6D). This result suggests that, apart from the major binding site, VAR2CSA features two other sites where CSA dodecamers can stably bind. These two sites are cryptic but get exposed upon opening of the core region of the VAR2CSA, due to the application of an external elongational force.

## Discussion

In this work, by using equilibrium and force probe MD simulations, we shed light on the molecular mechanism governing the increased VAR2CSA–CSA interaction under shear stress.

Our data demonstrate that upon force exertion, the core region of VAR2CSA undergoes a major conformational rearrangement by opening up in two structurally-intact subregions (Fig 3 and S1 Movie). The CSA molecule already bound at the major binding site operates as an anchor point, allowing for the opening of VAR2CSA to occur. This causes the subsequent exposure of two cryptic binding sites, which have the ability to accommodate further CSA molecules, prior to the dissociation of CSA from the major binding site (Fig 6). This is a remarkable result, as it proves that force causes neither the immediate dissociation of CSA from VAR2CSA nor its unfolding, but instead prepares the adhesin to bind further CSA chains (or other parts of the same chain) at two other locations. Consequently, our data support a mechanism in which the application of mechanical forces increases the valency of tethered VAR2CSA, and with it its avidity, to potentially reinforce its attachment to the proteoglycan matrix.

The recent low- [30] and high-resolution [21, 31, 34] structural information of VAR2CSA clarified the exact location of the minimal functional CSA binding unit around the DBL2X domain [27, 28] (i.e. the major binding site), but also suggested the possibility of a second CSA binding location, indicated by a single ASG monosaccharide observed in the holo cryo-EM structure of VAR2CSA [31](Fig 1). One of the cryptic binding sites exposed by force in our simulations precisely coincides with this proposed additional (minor) binding site (Fig 6). The other proposed binding site is precisely the region of the DBL4ϵ domain constitutive of the major binding site. When detached from the DBL2X domain, upon opening, this region had the ability to stably accommodate a dodecameric CSA chain by itself, while the protein was held by another sugar at the major binding site. Taken together, this reveals a key role of these two regions, namely the activation of additional CSA binding sites. Force, accordingly, controls the level of exposure of these sites.

Because of its dimensions ($\sim 17$ nm in the compact state), VAR2CSA is suggested not to be directly influenced by the hydrodynamic shear of the flowing blood. Instead, VAR2CSA is

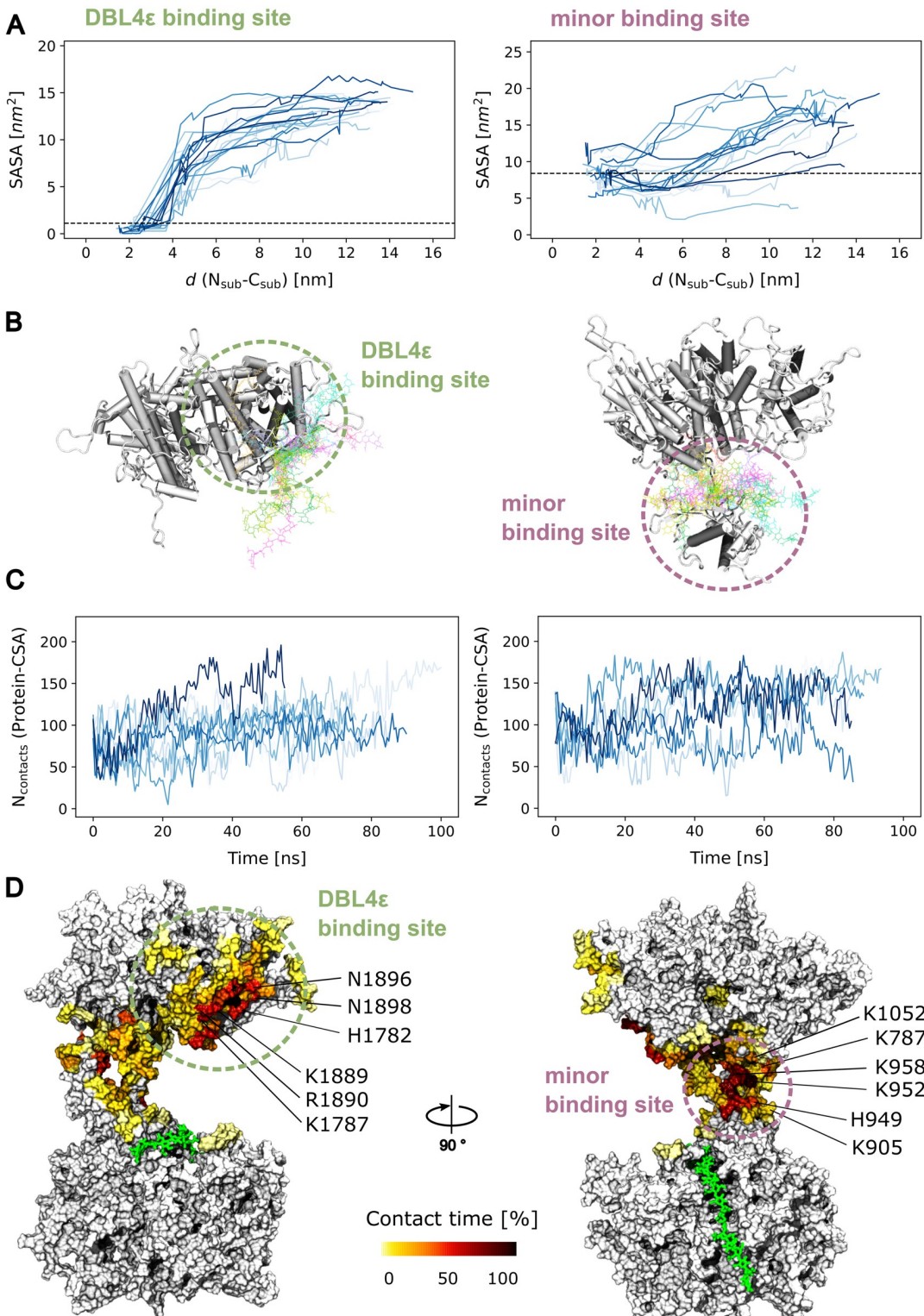

**Fig 6. Force exposes two cryptic secondary CSA sugar binding sites.** (A) Sugar accessible surface area (SASA) of the DBL4ε portion of the major binding site (left) and of the minor binding site at the interface between the DBL2X, ID2a, and DBL4ε domains (right) as a function of the opening distance between subregions $d(N_{sub} - C_{sub})$. The SASA was extracted from the pulling simulations (n = 20 for each case, different lines). For clarity, the SASA is shown every 10 ns for distances smaller than 4 nm and every 1 ns otherwise. The dashed line indicates the average SASA before opening (i.e. $t < 80$ ns). (B) Examples of the

conformations adopted by a CSA sugar dodecamer when it was docked and relaxed (by equilibrium MD simulations) at these two sites (sugar shown as coloured sticks and protein in cartoon). For clarity, only the relevant portion of the protein was shown. (C) Time-traces of the number of contacts established by the docked dodecamer and the protein during the MD relaxation simulations (n = 10; left: DBL4ε and right: minor binding site). (D) Contact time between protein residues at each identified binding site and the docked dodecamer were recovered from the MD relaxation simulations. The contact time is colour-coded on the surface of the protein, relative to the total time of the simulations (n = 8 amounting to ∼800 ns of cumulative simulation time). Positively charged residues displaying a large contact time are indicated. The glycan that remained bound to the DBL2X part of the major binding site is shown as green sticks.

assumed to experience an elongational tension when it is tethered to a CSA chain, at one side, while it is pulled by the infected erythrocyte by the action of the shear flow, at another side. Consequently, we applied a force along VAR2CSA that mimics such tension (Fig 3A and 3B). This resembles the behaviour of the blood clotting protein von Willebrand factor, which also becomes mechano-activated by the influence of an inhomogeneous tension distribution along the molecule generated by the shear of flowing blood [75, 76]. Although our choice is a reasonable assumption, in the future, additional simulations at a lower level of resolution but explicitly imposing a shear flow [77] would allow to directly establish the connection between elongational tension acting on VAR2CSA and the shear flow acting on the CSA-VAR2CSA-erythrocyte system.

We also investigated the mechanism that led to the conformational opening of VAR2CSA before the dissociation of the sugar from the major binding site (Fig 5). The direction of pulling strongly influences the force response of biomolecules [68–74]. Both unfolding of single protein domains [68–70, 72, 73] and dissociation of protein complexes [71, 74] exhibit kinetics that are dependent on the point of force application and the subsequent orientation of the molecule with respect to the force vector. Accordingly, pulling geometries in which several non-covalent stabilizing interactions need to simultaneously be broken offer greater mechanical resistance compared to geometries where they open sequentially one by one. This is not only a mere consequence of the experimental setup, but also a key feature biomolecules exploit to respond to force. They thereby act e.g. as force sensors [70], directed superstable bonds [71], or anisotropic switches [72, 73]. We suggest a similar effect applies here for VAR2CSA. The angle of the CSA–major binding site interface and the angle of the subregion–subregion interface with respect to the force vector suggest that these interfaces open through a "shearing" and "zipping" motion, respectively (Fig 5A and 5B). In competition, shearing of the sugar from the major binding site requires the rupture of multiple interactions at the same time. This is a mechanically more resistant and therefore a less likely process than the unzipping of the subregion interface. In addition, the interaction energy between the CSA chain and the major binding site was found to be stronger than that between the subregions, mainly due to the strong electrostatic attraction between the negatively-charged CSA moieties and the positively charged residues of the binding site (Fig 5C). This constitutes a difference between the CSA-VAR2CSA interaction and the interaction of neutral glycans with their ligands, for example between mannose and the *Escherichia coli* FimH adhesin [38, 78, 79]. While hydrogen bonds participate in both types of interactions, ionic pairs additionally contribute in the former case. Accordingly, the VAR2CSA–sugar interactions are highly force resistant due to, first, their orientation parallel to the force axis, which leads to a shearing mode for dissociation as opposed to the unzipping of the VAR2CSA core-subregions, and, second, the strong charge complementarity between the sugar and the protein major binding site. This dictates the preference for the opening and subsequent activation of VAR2CSA over its dissociation.

The observation of shear-enhanced adhesion for VAR2CSA [36] suggested that this adhesin employs a catch-bond mechanism, i.e. its adherence increases upon the application of force

[42]. Such behaviour is utilized by bacteria to attach to their glycan substrate [38] or by rolling leukocytes [39, 40]. It has also been proposed for another PfEMP1 variant [41]. However, locally, force did not seem to engage the CSA chain in a stronger interaction with the major binding site, as indicated by a loss of sugar–protein contacts upon force application (Fig 3D). Accordingly, we propose the major binding site of VAR2CSA does not strictly follow a catch-bond behaviour. Instead, force exposed two other sugar binding sites, which were cryptic under equilibrium conditions and where a CSA glycan could be stably maintained (Fig 6). Thus, VAR2CSA switches from a monovalent to a multivalent state by the action of force, thereby amplifying its avidity for CSA binding. Multivalency is a general mechanism to enhance ligand-receptor interactions [80] and a central feature for protein-glycan recognition [81]. We propose that VAR2CSA follows this principle to enhance its adhesion to the placental CSA matrix, but here controlled by the elongational tension acting on the protein as a result of the shear of flowing blood.

Our data emphasize the importance of conducting shear-binding assays [36], and not only static ones, to study VAR2CSA adherence. More specifically, our computational prediction of force control of the valency of VAR2CSA could be tested in such type of experiments. For instance, the opening of the core region could be artificially prevented, by introducing cross-linking cysteine residues at the N- and C-terminal subregions, and with this the shear-enhanced response is expected to be abrogated. The pairs of residues with highly affected mechanical stress upon opening, retrieved by FDA, constitute possible candidates for such cross-linking replacements.

## Conclusions

Based on equilibrium and force-probe MD simulations, we propose a mechanism for the inter-action of the *Plasmodium falciparum* malaria adhesin VAR2CSA with the placental CSA matrix. In this mechanism, mechanical force modulates the valency of tethered VAR2CSA, by opening of the core region in two structurally-folded subregions. Opening causes the exposure of two cryptic CSA binding sites. This enables VAR2CSA to stably bind a CSA chain at each of such exposed sites, in addition to the chain tethered to the major binding site. We attribute the preferential opening of the protein (and consequent exposure of cryptic binding sites), prior to the dissociation of CSA from the major binding site, to the orientation VAR2CSA adopts under tension and to the strong sugar–protein electrostatic interactions occurring at the major binding site. Rather than influencing the affinity of the major binding site, as it would be canonically thought for a catch-bond, controlling the valency of VAR2CSA by force is a possi-ble explanation of the observed shear-enhanced adherence of *Pf*-infected erythrocytes [36]. Accordingly, our *in silico* work introduces an interesting hypothesis into how VAR2CSA atta-ches more firmly *Pf*-infected erythrocytes to the proteoglycan matrix of the placenta, at the expense of the shear of the flowing blood. Overall, we propose, to our knowledge, a new mech-anism of adherence of *Pf*-infected erythrocytes to the placenta, which can be directly tested by experiments and which is relevant to the understanding of the malaria infection and to the development of vaccines against placental malaria targeting VAR2CSA.

## Supporting information

**S1 Fig. Intra-domain root mean square deviation (RMSD).** C$\alpha$ RMSD from the initial con-formation was computed separately for the nine different domains (panels) constituting the core region of VAR2CSA. Independent simulation replicas are shown in different colours (n = 10). (EPS)

**S2 Fig. Inter-domain contacts.** Time trace of number of contacts between domains, $N_{\text{contacts}}$, recovered from the equilibrium simulations of the core region of VAR2CSA. Each panel corresponds to one of the domain–domain pair, whereby the pairs that have no contacts over the entire simulation time are omitted. Each line corresponds to one simulation replica (n = 10, shades of blue). The red line indicates the mean of all replicas over the entire simulation time.
(EPS)

**S3 Fig. Force distribution analysis.** Change in pairwise force $\Delta F_{ij} = \langle F_{ij}(\text{open})\rangle - \langle F_{ij}(\text{closed})\rangle$ for the pair of residues $(i, j)$ for different threshold values of $z$ is shown. The representation is the same as the one used in the Fig 4 of the main text.
(EPS)

**S1 Movie. Force-induced opening of the VAR2CSA core region.** The protein is depicted in cartoon representation (gray and brown) and the CSA 21mer in sphere representation (green). Force was applied at the end of the CSA chain that prolongs to the CSA matrix and at the protein C-terminus.
(MP4)

## Acknowledgments

We thank Ulrich Schwarz, Motomu Tanaka, Michael Lanzer, Markus Ganter, and Manu Forero-Shelton for insightful discussions.

## Author Contributions

**Conceptualization:** Frauke Gräter, Camilo Aponte-Santamaría.

**Data curation:** Camilo Aponte-Santamaría.

**Formal analysis:** Nicholas Michelarakis, Frauke Gräter, Camilo Aponte-Santamaría.

**Funding acquisition:** Frauke Gräter.

**Investigation:** Rita Roessner, Nicholas Michelarakis, Frauke Gräter, Camilo Aponte-Santamaría.

**Methodology:** Rita Roessner, Nicholas Michelarakis, Camilo Aponte-Santamaría.

**Project administration:** Camilo Aponte-Santamaría.

**Supervision:** Frauke Gräter, Camilo Aponte-Santamaría.

**Validation:** Camilo Aponte-Santamaría.

**Visualization:** Rita Roessner, Nicholas Michelarakis, Camilo Aponte-Santamaría.

**Writing – original draft:** Rita Roessner, Nicholas Michelarakis, Frauke Gräter, Camilo Aponte-Santamaría.

**Writing – review & editing:** Rita Roessner, Nicholas Michelarakis, Frauke Gräter, Camilo Aponte-Santamaría.

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
