## [Decision Letter · Decision Letter 0]

28 Sep 2023

Dear Dr. Aponte-Santamaria,

Thank you very much for submitting your manuscript "Mechanical forces control the valency of the malaria adhesin VAR2CSA by exposing cryptic glycan binding sites" for consideration at PLOS Computational Biology. As with all papers reviewed by the journal, your manuscript was reviewed by members of the editorial board and by several independent reviewers. The reviewers appreciated the attention to an important topic. They were generally positive but had a set of recommendations to improve clarity and rigor, including some additional simulations and analyses.  Based on the reviews, we are likely to accept this manuscript for publication, providing that you modify the manuscript according to the review recommendations.

Sincerely,

Peter M Kasson

Academic Editor

PLOS Computational Biology

Nir Ben-Tal

Section Editor

PLOS Computational Biology

Reviewer's Responses to Questions

**Comments to the Authors:**

Reviewer #1: The authors analyze the molecular mechanism behind the shear enhanced activation of the binding between the VAR2CSA protein, which is produced by erythrocites infected with the malaria parasite Plasmodium falciparum, and the CSA receptor located on the placenta of pregnant women. The results suggest a catch-bond-like mechanism, but instead of increasing the life time of the bond between protein and ligand tensile force provides an additional two minor binding sites for the CSA receptor, which consists mainly of a dodecameric chain, hence increasing the valency of the protein-receptor interaction. This mechanism is supported by the observation in force probe simulations that the major binding site in VAR2CSA opens but the ligand remains bound. Such opening exposes two additional cryptic binding sites, which the authors show are large enough to accommodate each one additional CSA dodecameric glycan chain. The authors use a combination of equilibrium simulations (with no external tensile force) to test the stability of the VAR2CSA structure, which also contained segments that the authors had to model, force probe simulations with pulling at constant velocity, and computational docking where dodecameric glycan chains were docked to the cryptic sites that became exposed in the force probe simulations. The docking was then also followed by equilibrium simulations. The authors also performed various analysis, such as the number of contacts between protein and ligand along the simulation, and energetic calculations using the force field terms. Overall, the work is well done and will be of interest to the biophysical community.

I have a few minor comments, which are aimed at improving clarity and adding more quantification to the analysis:

1) Calculation of contacts: In the methods it says that the tool gmx mindist was used. I do not understand between what the minimum distance is calculated. Is it the minimum distance between two residues or the distance between two atoms? What is the distance cutoff used to define that two residues/atoms are forming a contact?

2) In Figure 3D the claim is that the number of contacts decreases with opening distance and with simulation time. This is very difficult to tell by just looking at the plot. I think that the authors should calculate the Pearson's linear correlation coefficient, at least between number of contacts and opening distance. (Hint: in case the authors use xmgrace they may be aware that this can be done with: Data -> Transformations -> Regression)

3) Bottom paragraph of page 11 discusses energetic calculations and the Coulombic attraction between the binding site and the glycan chain. Have the authors performed a hydrogen bond analysis between VAR2CSA and the glycan? For example, there are hydrogen bonds between the adhesin FimH (located at the tip of bacterial fimbria) and mannose, which are important for the stability of the bound conformation. I suspect that the hydrogen bonds between ligand and protein can provide a physical explanation for the larger Coulombic attraction between ligand and protein than between the two sub-regions. Can the authors identify examples of key persistent hydrogen bonds between VAR2CSA and CSA (major binding site) in the 200-ns long equilibrium simulations? Often, a contact is defined as persistent when it is formed in at least 66% of the simulation frames (excluding, e.g., the first 10 ns or so until the system has equilibrated), but it is up to the authors what criteria or cutoffs are best suited.

Other minor comments:

4) Bottom of page 2 Introduction: "They are connected by four complex inter-domain regions (ID1 - ID3) ...". The list looks like it is only three. They should be all listed.

5) Similarly, "... six Duffy-like binding domains (DBL1 - DBL6) ...", but then it talks about "DBL2X" and "DBL4epsilon". I think they should be listed with their names when they are introduced.

6) Page 6 sixth line: "domain portion of the major binding site that got exposed ..." This becomes clear after reading the Results, but it is a bit confusing. The authors may want to write: "domain portion of the major binding site that is observed to become exposed in the force probe simulations".

7) At the end of the first paragraph at page 6, I had to read it many times to understand this part of the sentence: "..., which was not zero when their distance got below a distance of 13 nm". Maybe something like "..., which was non-zero when their distance was below 13 nm".

8) In section "Force opens the core region of tethered VAR2CSA", I think that it is not really necessary to repeat the methods such as the speed of 0.2 m/s and that "20 conformations were extracted from the equilibrium MD simulations". This information can probably be shortened while referring the reader to the Methods for details.

9) In "Conclusions", I got confused how many ligands in total were docked to the minor (cryptic) sites. It states: "Opening causes the exposure of two cryptic CSA binding sites. This enables VAR2CSA to stably bind a second CSA chain in addition to the one tethered to the major binding site". Is one additional CSA chain bound at the same time to both cryptic sites, or should it read: "This enables VAR2CSA to stably bind two additional CSA chains"?

10) Last sentence of "Conclusions" should read (my additions or changes are in square brackets) "..., which can [be] directly tested by experiments and which may be relevant [to] the understanding of the malaria infection and [to] the development ...".

11) Second sentence of Author summary: "Malaria is caused [by] infection [with] the Plasmodium parasite, ..."

12) Page 3 third paragraph, it should be "fimbrial adhesin" instead of "fimbrian adhesin".

13) "helped VAR2CSA maintain" instead of "maintaining".

14) The article needs to be proofread for other misspellings and typos.

Reviewer #2: In the manuscript titled “Mechanical forces control the valency of the malaria adhesin VAR2CSA by exposing cryptic glycan binding sites”, Roessner et al. delve into the molecular mechanism governing the enhanced adhesion of Plasmodium falciparum-infected erythrocytes to the placenta under shear flow conditions. The adhesion process is coordinated by the VAR2CSA protein, contributing significantly to pregnancy-associated malaria. The authors employ a set of simulation methodologies to reveal how mechanical forces can instigate significant conformational changes in VAR2CSA, unveiling additional binding sites and increasing its binding affinity from a monovalent to a multivalent state. The insights gathered from this study lay the foundation for the development of targeted vaccines against placental malaria.

Overall, the manuscript is well written, and the simulations employ cutting-edge methodologies. Constructing the structure of this complex was monumental and deserves acknowledgment as a significant achievement, meritorious of publication in its own right. The system, comprising approximately 700,000 atoms, underwent combined simulations of 2 microseconds, utilizing equilibrium molecular dynamics simulations. This facilitated a thorough examination of the structure's stability.

However, I have a few reservations regarding the stability of the structural model:

1. The increasing number of contacts over time in Figure 2D might indicates that the model structure hasn’t reached stability and might not be in its native conformation.

2. The high RMSD in Figure 2C could be masking large scale instability in some domains.

To address these concerns, I suggest performing an RMSD analysis for each protein domain independently, allowing for an in-depth assessment of the stability of individual domains. This approach could also pinpoint which sections of the protein complex are experiencing greater motion, thereby identifying potential areas of instability. Also, the authors could use the same approach to check two protein domains at the time, helping to identify motion between protein domain pairs.

Additionally, the manuscript includes a mechanical study utilizing steered molecular dynamics simulations. The size of the system and the number of replicas are admirable. However, I would like to propose an enhancement:

The discussion surrounding the analysis in Figure 3, especially 3D and 3E, could gain depth with the inclusion of a force propagation analysis. Given that one of the manuscript's authors has previously developed strategies for such analyses, incorporating this would provide a richer understanding of the geometric aspects of the mechanical bond, currently best represented by Figure 4A.

Furthermore, the discussion contrasting zipping versus shearing motion deserves more comprehensive coverage within the paper. This mechanism appears to be a pivotal finding and thus, a more elaborate discussion would greatly enrich the readers’ understanding. This is especially pertinent as this mechanism has been identified in various other systems. The difference between both zippering and shearing has been also identified within a single protein complex by examination of how tethering geometry impacts unbinding mechanics.

In closing, I am a great fan of the work, and I hope that my suggestions help to improve it even further.

**Have the authors made all data and (if applicable) computational code underlying the findings in their manuscript fully available?**

Reviewer #1: Yes

Reviewer #2: Yes

PLOS authors have the option to publish the peer review history of their article (what does this mean?). If published, this will include your full peer review and any attached files.

Reviewer #1: No

Reviewer #2: No

Figure Files:

Data Requirements:

Reproducibility:

References:

---

## [Editor Report · Decision Letter 1]

2 Dec 2023

Dear Dr. Aponte-Santamaria,

We are pleased to inform you that your manuscript 'Mechanical forces control the valency of the malaria adhesin VAR2CSA by exposing cryptic glycan binding sites' has been provisionally accepted for publication in PLOS Computational Biology.

Upon reviewing your responses to reviewer requests and modifications to the manuscript, I think you have done an excellent job of responding to the reviews.  Since the reviewers were positive in the first instance, I find another round of peer review is not required.

Best regards,

Peter M Kasson

Academic Editor

PLOS Computational Biology

Nir Ben-Tal

Section Editor

PLOS Computational Biology

---

## [Editor Report · Acceptance letter]

11 Dec 2023

PCOMPBIOL-D-23-01231R1 

Mechanical forces control the valency of the malaria adhesin VAR2CSA by exposing cryptic glycan binding sites

Dear Dr Aponte-Santamaría,

I am pleased to inform you that your manuscript has been formally accepted for publication in PLOS Computational Biology. Your manuscript is now with our production department and you will be notified of the publication date in due course.

With kind regards,

Anita Estes
